# A Novel Microwave Treatment for Sleep Disorders and Classification of Sleep Stages Using Multi-Scale Entropy

**DOI:** 10.3390/e22030347

**Published:** 2020-03-17

**Authors:** Daoshuang Geng, Daoguo Yang, Miao Cai, Lixia Zheng

**Affiliations:** 1School of Mechanical and Electrical Engineering, Guilin University of Electronic Technology, Guilin 541004, China; emegyd@guet.edu.cn (D.G.); caimiao105@163.com (M.C.); 2College of Continuing Education, Guilin University of Electronic Technology, Guilin 541004, China; lixia_zhengguet@163.com

**Keywords:** sleep disorders, sleep stage, microwave scattering, entropy features, brain activity

## Abstract

The aim of this study was to develop an integrated system of non-contact sleep stage detection and sleep disorder treatment for health monitoring. Hence, a method of brain activity detection based on microwave scattering technology instead of scalp electroencephalogram was developed to evaluate the sleep stage. First, microwaves at a specific frequency were used to penetrate the functional sites of the brain in patients with sleep disorders to change the firing frequency of the activated areas of the brain and analyze and evaluate statistically the effects on sleep improvement. Then, a wavelet packet algorithm was used to decompose the microwave transmission signal, the refined composite multiscale sample entropy, the refined composite multiscale fluctuation-based dispersion entropy and multivariate multiscale weighted permutation entropy were obtained as features from the wavelet packet coefficient. Finally, the mutual information-principal component analysis feature selection method was used to optimize the feature set and random forest was used to classify and evaluate the sleep stage. The results show that after four times of microwave modulation treatment, sleep efficiency improved continuously, the overall maintenance was above 80%, and the insomnia rate was reduced gradually. The overall classification accuracy of the four sleep stages was 86.4%. The results indicate that the microwaves with a certain frequency can treat sleep disorders and detect abnormal brain activity. Therefore, the microwave scattering method is of great significance in the development of a new brain disease treatment, diagnosis and clinical application system.

## 1. Introduction

Over the years, sleep disorder has become one of the most serious public health problems worldwide because it affects the physical health and mental state of individuals. Sleep is a very common physiological phenomenon in the daily life of human beings and animals. Good sleep quality can ensure a high-quality living state and spiritual vitality. Sleep disorders can be found everywhere, for instance, in Japan, surveys show that 29% of people sleep less than six hours per day, 23% report insufficient sleep, 6% take hypnotics, 21% have underlying symptoms of insomnia, and 15% are excessively sleepy during the day [1]. During sleep, consciousness and brain activity undergo unusual changes within a very short period, with sleep signals varying in strength and intensity. Simultaneously, the fast, low-amplitude desynchronized electroencephalographic activity of wake is replaced by high-amplitude slow waves and spindles during sleep [2]. Animals in the process of sleep undergo mainly anabolism to restore physical strength and energy [3]. Similarly to eating, breathing, and walking, sleep plays an important role in regulating function and physical recovery, such as reorganizing the cortex associated with learning and memory, and processing information to consolidate memories [4]. Sleep also helps the body and brain repair cells and maintain the functional integrity of the immune system [5]. A good night’s sleep can also boost immunity and help prevent contraction of the currently prevalent novel coronavirus pneumonia (COVID-19).

Sleep is important to people and thus, the treatment of sleep disorders remains a research focus. Most people with sleep disorders are treated with drugs such as 5-hydroxytryptamine and acetylcholine, which could cause potential harm to the body over time and can have a rebound effect [6]. The effect of treatment is obvious initially, then, after a time, the drug gradually appears to "fail" [7]. Many new forms of physical therapy, such as using sound, light, electricity, and magnetism to act on certain parts of the body [6,7,8,9], have been used to induce sleep. Electricity and magnetism can also stimulate neurons in the functional areas of the brain; a special frequency of weak electricity and weak magnetic field causes the resonance phenomena to affect the discharge frequency of brain functional areas, which, in turn, induces sleep [10]. However, the equipment used is relatively expensive and bulky and currently exists in laboratories and rarely used in daily life. A considerable amount of experimental data have proven that the brain’s sleep can be improved through external conditions. For example, neural drugs can be used to stimulate the brains of insomniacs and suppress neuronal activity to achieve the effect of sleep [11]. Intense exercise is also believed to improve the quality of sleep [12]. However, despite previous efforts, the growing number of people suffering from insomnia indicates the solutions are flawed. Drugs can damage people’s health and exercise is not suitable for people with disabilities.

Sleep stage detection and recognition is an important basis for measuring and evaluating sleep quality. Sleep stage detection and recognition is usually conducted with polysomnography (PSG), including electroencephalogram (EEG), electrooculogram (EOG), electromyogram (EMG), electrocardiogram (ECG), chest and abdomen movements, pulse oxygenation, and snoring [13,14]. These technologies involve the use of a considerable number electrodes, which is very inconvenient for the measurement and treatment of sleep disorders and may bring physical and psychological interference to patients, and even affect the quality of sleep [15]. The microwave scattering technique is a new method for testing brain activity. Microwaves can be used to detect brain activity such as sleep, pain perception, epilepsy, depression, and motor imagery [16,17,18], and it has been shown to be capable of treating insomnia and other brain diseases [19]. This technique is a non-contact test method to avoid the inconvenience and anxiety caused by electrode attachment to the scalp and to control the contamination of brain activity by physiological signal artifacts (i.e., EMG, EOG) [17,18]. Depending on its ability to record brain activity, it can also be used in the study of sleep stage [19]. Compared to existing techniques, the EEG is divided into invasive and non-invasive types. Invasive EEG causes irreversible damage to the brain, whereas non-invasive EEG (scalp EEG) has very low spatial resolution [5]. The use of EMG and EOG in detecting the sleep has undergone considerable progress but these methods rely on too many physiological traits and physiological response in the time dimension tends to be delayed by 5–6 s, which limits the time dynamics of the brain’s stimulus response; the low temporal and spatial resolution may also ignore some important information in the signal [13,14].

In terms of brain function detection, microwave has attracted extensive attention because of its low cost, high contrast imaging, its non-destructive and non-invasive advantages, as well as the fact that it is not limited by temporal resolution and spatial resolution [19]. The early application of microwave scattering technology has mainly been for the detection of cerebral strokes and brain tumors. Mobashsher et al. [20,21] successfully improved a simple microwave detection system, designed a realistic human head model, imaged the status of intracranial hemorrhage, and developed a method for locating and analyzing the bleeding target according to the frequency scattering characteristics of the bleeding site. Kandadai et al. [22] developed a microwave frequency scanning monitoring device and used the microwave transmission method to analyze the changes in the strength of the signal of pig brain edema. Zamani and colleagues [23] estimated the scattering power intensity in the imaging region in real-time from measured multi-static microwave scattering signals outside the brain functional imaging region. These results indicate that the scattering of microwave in the brain functional sites leads to the polarization and depolarization of the brain tissue cell fluid, resulting in the change in dielectric constant and conductivity, which in turn changes the characteristics of microwave scattering to achieve the purpose of imaging [19]. Some recent reports have addressed the fact that microwave scattering has a significant effect on resting state EEG, particularly on the fact that the power of the EEG increases in the alpha band [24,25]. Exposure to electromagnetic fields can cause changes in the sleep EEG power [26], thereby suggesting that electromagnetic waves may affect brain function and the recovery process involved in sleep.

Sleep states need to be classified effectively and automatically to measure sleep quality more efficiently. For the classification of sleep stages, one of the most persistent problems is identifying and distinguishing the features, called feature extraction [27]. However, determining which algorithms can identify the valid features of a given problem is a very complex activity. Traditional insomnia stages use PSG to extract power, energy spectra, autoregressive models, and multiscale entropy using EEG, EMG, and EOG signals [28,29]. Then, the classifier is combined with the linear discriminant method, random forest (RF), support vector machine (SVM), Naive Bayes (NB), Sleep Stage Transformation (SST) model, and Time-Varying Sleep Stage Transformation (TSST) model [30,31]. However, these methods are computationally complex, with some requiring 41 features to be extracted from EEG and physiological signals. These models are also accurate enough to meet clinical needs [32,33,34]. The microwave modulation and detection technology proposed in this paper can avoid the physiological artifact of patients and can improve successfully the recognition accuracy of sleep stages by combining with the multiscale entropy features that can better distinguish different types of information. Some reports indicate that low-frequency microwave modulation frequency can alter the variations of EEG power in the resting state [35]. Other reports have shown that microwave frequency modulation can affect sleep and that changing the frequency modulation can prolong or shorten sleep time [36]. Li et al. [19] determined that microwaves can affect sleep and wakefulness in anesthetized rats using 30 GHz radiofrequency electromagnetic radiation and confirmed that microwaves have a higher spatial resolution than EEG. Using the principle of microwave scattering, Wang et al., [16] proved that microwave transmission signals could represent the dynamic characteristics of the brain in motion imagination by adjusting the transmission position of the antenna. Compared with EEG, fMRI, and other imaging methods, the microwave method can obtain purer brain activity signals. Some physiological artifacts are also avoided because the antenna is not in contact with human skin. Geng et al. successfully realized the recognition and classification of different pain categories by analyzing the microwave transmission signals. Their classification accuracy reached more than 90%, higher than that of most existing methods for pain classification [17,18].

Quantifying the dynamic irregularity of time series is an important challenge in signal processing. Entropy is an effective and extensive method for measuring the irregularity and uncertainty of time series [32]. The complexity of the time series characterized by entropy value shows different trends with the increase of the time scale. The greater the entropy, the greater the uncertainty; the higher entropy means higher uncertainty and lower entropy means lower irregularity or uncertainty [37,38]. Time series recorded by dynamic physiological systems usually show long-term correlation on multiple time scales; hence, time series related to brain activity have multiple and synchronous activity mechanisms that usually span multiple time scales [39,40,41,42,43,44]. Therefore, because entropy cannot describe brain dynamics fully on a single time scale, multiscale entropy is introduced to quantify the complexity of the system [42,45,46]. The entropy estimation result of the sample entropy (SampEn) algorithm is not always related to the complexity. Costa et al. introduced the multiscale (sample) entropy (MSE) to express the complexity of time series [40,41]. MSE algorithm solves the contradiction of low entropy and high complexity between 1/f noise and white noise. However, MSE may not be able to obtain accurate sampEn in large-scale coarse-grained time series [30]. Wu and his colleagues [39] sought to counter the problem of MSE algorithm accuracy by proposing a better overall performance of refined composite multi-scale sample entropy (RCMSE). Multiscale fluctuation-based dispersion entropy (MFDE) is based on the dispersion entropy (DispEn); it introduced one type of measured approach to the time series of entropy uncertainty [42] and is the quantitative multiple time scales of new methods of physical dynamics. MFDE avoids the problem of undefined MSE value and makes the entropy of white noise and 1/f noise to become more stable than the scale factor. However, MFDE ignores the relative frequency of each fluctuation-based dispersion pattern of shifted series [40,41]. Multiscale fluctuation-based dispersion entropy (RCMFDE), which is defined as the shift sequences based on the fluctuation of the average rate of dispersion pattern of Shannon entropy, has been introduced to overcome the abovementioned problem and distinguish between the different neural data status in the fastest and most consistent manner [42]. Multivariate multiscale weighted permutation entropy (MMSWPE) is also a method for measuring the dynamic complexity of brain activity. Quantifying the regularity of time series on a single time scale based on the traditional permutation entropy (PE) may lead to false results of nonlinear time series [38]. Hence, MMSWPE combines weighted permutation entropy and multivariate multiscale method to quantify the characteristics of different brain regions and multi-time scales as well as the amplitude information contained in multi-channel EEG signals [37]. While using multiscale entropy to measure the complexity of time series, focus should also be given to the choice of the appropriate scale for entropy calculation. For example, a too small-scale selection is likely to result in insignificant features. When the scale is too large, the calculation is complicated and distinguishing the complexity index of different time series is easy. In addition, entropy is usually used to characterize biomedical signals and cause an improvement in the classification accuracy and recognition efficiency. Geng et al. used the multi-scale entropy feature combined with SVM-RF classifiers to classify and evaluate different pain types with an accuracy of more than 93% [18]. Rahman et al. [47] used statistical features, such as spectral entropy and refined composite multiscale dispersion entropy (RCMDE) in the discrete wavelet transform (DWT) domain analysis of single-channel EOG signals and used RUSBoost classifier for automatic sleep stage classification with an average accuracy of 84.70%. Liang et al. [48] used the multiscale entropy method to process EEG signals and linear discriminant analysis to conduct automatic sleep staging with an average accuracy of 76.91%. Tian et al. [49] combined multi-scale entropy characteristics with the proportion information of sleep structure and proposed a hierarchical sleep automatic scoring method. The multi-scale entropy (MSE) was extracted from EEG to characterize the signal characteristics at multiple time scales while the SVM was used to achieve an accuracy of 85.60%. Therefore, the use of entropy as a signal feature to characterize the sleep-related time series may improve the classification accuracy of sleep stages.

We improved the sleep experiment scheme based on previous research experience to improve the observability and controllability of sleep experiments. Based on the principle of microwave scattering and using microwave modulation and detection technology, the satisfaction score of sleep quality and frequency band energy statistics of patients with sleep disorders were analyzed to evaluate the improvement in their sleep quality. The effects of microwave modulation on the treatment of sleep disorders and the accuracy of sleep staging were also tested. The detailed operation process is as follows. First, the specific frequency of microwave was used to penetrate the functional sites of the brain in patients with sleep disorders and the firing frequency of the sleeping brain activity was changed to realize the detection and treatment of sleep stages. Secondly, the wavelet packet algorithm was used to extract fine composite samples of RCMSE, RCMFDE, and MMSWPE as a feature data. Thirdly, the feature selection method based on mutual information-principal component analysis (MIPCA) was used to optimize the feature data set and the feature selection algorithm was used to identify the features providing the highest effect. Finally, the RF classifier was used to organize and evaluate sleep stages. 

## 2. Experimental Design and Methods

### 2.1. Experimental Design and Data Collection

The twenty-four male volunteers with sleep disorders were all from Guilin University of Electronic Technology (GUET), right-handed with an average age of 22.65 years (age range of 21 to 25 years). All the volunteers had no neurological or physical problems other than sleep disorders, no long-term use of other psychiatric drugs or brain damage, and no alcohol, tea or coffee consumption during the sleep experiment. Before the experiment began, all volunteers were labeled as S1–S24. This study was approved by the ethics committee of the Guilin University of Electronic Technology and complies with the relevant laws of China and conducted in conformity with the Helsinki declaration.

The test was conducted in two rooms with good sound insulation, dim light, and no strong magnetic and electric interference. The indoor temperature was controlled at about 25 °C. The layout of the two rooms is exactly the same, with each room having a bed of the same arrangement (complete bedding). The brain functional site that regulate sleep is the ventrolateral preoptic nucleus (VLPO), which is about 1 cm in size. Hence, the wavelength of 6 GHz electromagnetic wave in vacuum was 6 cm to ensure that the near-field of the antenna can cover the functional sites of the brain to achieve higher detection and modulation efficiency. The near-field of the electromagnetic wave is formed through the antenna. ζ refers to the distance from the antenna to be tested to the boundary of the near field, D is the maximum size of the antenna’s physical diameter, and λ is the wavelength of an electromagnetic wave in a vacuum. The three conditions should be satisfied as follows:(1)$$0<\zeta <0.62\times \sqrt{\frac{D^{3}}{\lambda }}.$$

Therefore, for the regulation and treatment of sleep disorders, the distance of a microwave antenna from VLPO’s brain functional sites is $0<\zeta <0.62\times \sqrt{\frac{12^{3}}{6}}\approx 10.5$.

This study is composed of two procedures, namely, sleep treatment detection and classification of sleep stages. The experiment used the recording of PSG combined with the microwave transmission signal, in which PSG recorded the sleep stage activities and microwave transmission signals were used to replace EEG or EOG for the classification of sleep stages. The PSG recordings were obtained by utilizing the Jaeger-Toennies system (the sample rate was 256 Hz). The PSG recordings for each subject included six EEG channels (F3–A2, F4–A1, C3–A2, C4–A1, P3–A2, and P4–A1, according to the international 10–20 standard system), two EOG channels, and one EMG channel. The microwave transmission signal record is a column of microwave phase-changing data. Ag/AgCl alloy electrodes were used to collect physiological signals and the electrode impedance is less than 5000 Ω. The electromagnetic wave receiving equipment is a pair of wideband horn antenna (ChengDu Ainfo Inc., Chengdu, China), with a frequency of 2.0–18.0 GHz, transmission gain of 12 dB, and standing wave of 2.0:1. The equipment is also capable of withstanding the maximum continuous wave power of 50 W. The electromagnetic wave generation and signal processing equipment is an Agilent two-port vector network analyzer (Agilent Technologies N5230A; Agilent Technologies, Inc. USA) with a receiver measurement sensitivity of −120 dB, measurement frequency range of 300 K–20 GHz, and trace noise of 0.005 dB (when the intermediate frequency width is 10 kHz). For microwave modulation and detection, the sampling frequency is 250 Hz, the microwave transmission frequency is set to 6 GHz, and the microwave modulation frequency is set to 20 Hz. The transmission coefficient was S21 (S-parameters). Each night, two subjects were placed in one of the beds as they normally sleep. The test with microwave propagation is called a true machine test while the test without microwave propagation is called a false machine test. The microwave test process is shown in Figure 1. After the experiment began, none of the participants were allowed to use their phones or do anything other than sleep. Each test lasted 480 minutes and each person was subjected to eight tests, including four true machine tests and four false machine tests. The experiment began at 0:00 p.m. and ended at 8:00 a.m. After each experiment, data from the detector tested by the true machine was transmitted to the computer through the Bluetooth device for later data processing, and then the zeroing detection device carried out the next test. At the end of each test, each participant filled out a survey rating form that measure their rate of satisfaction with sleep improvement. In this table, Numbers from 0 to 10 are used to represent different satisfaction scores. The scoring criteria are as follows: 0 = very dissatisfied, 1–3 = dissatisfied, 3–5= slightly dissatisfied, 6–7 = slightly satisfied, 8–9 = relatively satisfied, 10 = very satisfied. 

### 2.2. Data Preprocessing

In this study, only microwave transmission signals were analyzed, and the EMG and EOG were used only as references to determine sleep stages. Prior to data analysis, the acquisition of microwave transmission signal was digitally bandpass filtered using a fourth-order Butterworth filter between 0.5 and 150 Hz. The forward and backward filtering is used to reduce phase distortion. Finally, the time series was digitally filtered using the hamming window FIR bandpass filters of order 200, with cutoff frequencies of 0.5 Hz and 40 Hz, which are usually used to analyze brain activity. Brain activity signals within the range of 0.5–30 Hz are generally a focus of concern for clinical medical research because of the wide frequency span of microwave transmission signals. The results of microwave transmission signals and denoising before and after are shown in Figure 2. According to the R&K sleep staging criteria, sleep is divided mainly into wake (W), non-rapid eye movement (NREM), and REM. The NREM stage is divided into NREM stage I (S1), NREM stage II (S2), NREM stage III (S3), and NREM stage IV (S4). According to the standards of the American Academy of Sleep Medicine in 2007, S1 and S2 are combined into light sleep (LS), while S3 and S4 are combined into slow-wave sleep (SWS). Here, the rhythm information contained in different sleep stages is as follows: the alpha wave (8–13 Hz) and the beta wave (13–30 Hz) in stage W; the theta wave (4–8 Hz) is the main manifestation in the stage LS, K-complex (0.5–1.5 Hz), and sleep spindle (12–14 Hz) are also observed; delta wave (0.5–4 Hz) is the main manifestation in the stage SWS; in stage REM, it is represented mainly by alpha, beta and theta waves, as well as some sawtooth waves (2–6 Hz).

### 2.3. Sleep Quality Measurement and Statistics

We verify whether the 6 GHz microwave can detect and treat the abnormal brain activity of patients with sleep disorders and compare the energy changes of the output data of different stages. The significance level *p* ≤ 0.05 has statistical significance. Data calculation was performed in MATLAB (2014b)^®^ (The Math Works Inc., Natick, MA, USA). Data statistics and analysis were implemented using SPSS 22.0^®^ (IBM Corp., Armonk, NY, USA). The sleep quality evaluation was based on the sleep standard of normal adults. Sleep onset latency (SOL) refers to the time between the patient’s waking state and stage S1, which is generally short, accounting for about 5% of the total sleep time in normal people. The number and duration of sleep awakenings are the total number of periods of W that occur during sleep. Total sleep time (TST) is the total amount of sleep. The proportion of sleep time (also known as sleep efficiency) refers to the ratio of TST to time in bed (TIB), that is, sleep efficiency = TST/TIB, which is generally more than 80% is considered normal sleep. The sleep maintenance rate (SMR) is the ratio of the TST to the time the person falls asleep to the time they wake up in the morning; generally, a score of more than 90% is normal. In addition, it is normal for NREM to account for more than 75%–80% of sleep time, in which S1 accounts for 2%–5%, S2 for 45%–55%, S3 for 3%–8%, and S4 for 10%–15%. REM sleep accounts for 20%–25% of TIB. The paired *t*-test was used to analyze the hypnotic effect of sleep disorder patients in a specific microwave frequency band.

### 2.4. Sleep Stage Recognition and Feature Extraction

After statistical analysis, we verify the accuracy of microwave detection of sleep stages by extracting features of microwave transmission signals for classification and recognition. Forty groups (each 10 s length) of data were selected randomly from each stage without repeating for feature research because of the very large experimental data sampling frequency (250 Hz), long sampling time (28,800 s), and long signal length of each trial (7.2 million data points). We used the time window function, taking 10 s data length as the basic unit of the time series. Wavelet packet transform was used to decompose each time series by five levels. The mother wave was Daubechies-4, and 32 wavelet packet coefficients (that is, 32 frequency bands) were obtained and reconstructed. The frequency bandwidth of each coefficient is 125/32 ≈ 3.9 Hz. A(5,0), D(5,1), D(5,3), and D(5,2)+D(3,1) were selected from the 32 wavelet packet reconstruction coefficients. The corresponding frequency bands were 0–3.9 Hz, 3.9–7.8 Hz, 7.8–11.7 Hz, 11.7–15.6 Hz, and 15.6–31.25 Hz, corresponding to delta wave, theta wave, alpha wave, and beta wave, respectively. The entropy coefficients of RCMSE, RCMFDE, and MMSWPE were extracted from these nodes as features [37,38]. The detailed solution methods for the three entropy features are as follows.

#### 2.4.1. RCMSE Extraction

In this study, the calculated entropy is the RCMSE based on the mean value. For a univariate signal of length *L*: $x=\left\{x_{1},x_{2},\cdots ,x_{L}\right\}$. Hence, to compute RCMSE, we have to solve for MSE. Solving MSE involves two steps [39]: (a) the coarse-grained process is used to obtain the representations of the original time series on different time scales and (b) Sample entropy (SampEn) is used to quantify the regularity of coarse-grained time series. For a given sampling power *n* and tolerance *r*, let $n^{m}$ represent the total number of *m*-dimensional matching vector pairs, and get $n^{m+1}$ to represent the total number of (*m* + 1)-dimensional matching vector pairs. SampEn is defined as the logarithm of the ratio of $n^{m+1}$ to $n^{m}$, as follows:(2)$$SampEn(x,m,r,\tau )=-\ln\frac{n^{m+1}}{n^{m}},$$
where *m* is the embedded dimension, *r* is tolerance, and *τ* is the scale factor.

The length of the original time series is divided into a non-overlapping window for *τ* and for an average of data points in each window to obtain the scale factor for *τ* coarse-grained time series. The *k*-th coarse-grained time series $y_{k}^{\left(\tau \right)}=\{y_{k,1}^{\left(\tau \right)},y_{k,2}^{\left(\tau \right)},\cdots ,y_{k,L}^{\left(\tau \right)}\}$ of x is defined as follows [40]:(3)$$y_{k,j}^{\left(\tau \right)}=\frac{1}{\tau }\sum_{i=(j-1)\tau +k}^{j\tau +k-1}x_{i},$$
where $1\leq j\leq \left\lfloor \frac{L}{\tau }\right\rfloor =N$, 1≤k≤τ, which means the coarse-grained series are calculated as the mean of a continuous sample [41]. *τ*, the scale factor of the MSE, is defined as the first coarse-grained SampEn time sequence, as follows:(4)$$MSE(x,m,r,d,\tau )=SampEn(y_{1}^{\left(\tau \right)},m,r).$$

In the case of the scaling factor for τ, calculate all the coarse-grained sequence matching vector of $n_{k,\tau }^{m+1}$ and $n_{k,\tau }^{m}$. Let ${\bar{n}}_{k,\tau }^{m+1}$ or ${\bar{n}}_{k,\tau }^{m}$ represent the average value of $n_{k,\tau }^{m+1}$ or $n_{k,\tau }^{m}$ in the range 1≤k≤τ. The RCMSE value of the scale factor τ is defined as the logarithm of the ratio of ${\bar{n}}_{k,\tau }^{m+1}$ to ${\bar{n}}_{k,\tau }^{m}$. That is:(5)$$RCMSE(x,m,r,d,\tau )=-\ln\frac{{\bar{n}}_{k,\tau }^{m+1}}{{\bar{n}}_{k,\tau }^{m}},$$
where ${\bar{n}}_{k,\tau }^{m+1}=\frac{1}{\tau }\sum_{k=1}^{\tau }n_{k,\tau }^{m+1}$, ${\bar{n}}_{k,\tau }^{m}=\frac{1}{\tau }\sum_{k=1}^{\tau }n_{k,\tau }^{m}$, *d* is the delay time. Therefore, Equation (5) can be simplified as:(6)$$RCMSE(x,m,r,d,\tau )=-\ln\frac{{\bar{n}}_{k,\tau }^{m+1}}{{\bar{n}}_{k,\tau }^{m}}-\ln\frac{\frac{1}{\tau }\sum_{k=1}^{\tau }n_{k,\tau }^{m+1}}{\frac{1}{\tau }\sum_{k=1}^{\tau }n_{k,\tau }^{m}}=-\ln\frac{\sum_{k=1}^{\tau }n_{k,\tau }^{m+1}}{\sum_{k=1}^{\tau }n_{k,\tau }^{m}}.$$

Therefore, it can be concluded that RCMSE values are undefinable only when both $n_{k,\tau }^{m+1}$ and $n_{k,\tau }^{m}$ are zero and should be avoided when calculating RCMSE.

#### 2.4.2. RCMFDE Extraction

Given a time series of length L: $x=\left\{x_{1},x_{2},\cdots ,x_{L}\right\}$, solving RCMFDE to calculate fluctuation-based dispersion entropy (FDispEn) and MFDE. The signal is mapped to *c* classes with integer indices from 1 to *c*. The original signal is divided into non-overlapping segments of length, known as scale factors τ. The average value of each segment to get a coarse-grained time series can be calcultated as shown below [42,43,44]:(7)$$y_{j}^{\left(\tau \right)}=\frac{1}{\tau }\sum_{b=(j-1)\tau +1}^{j\tau }x_{b},\;1\leq j\leq \left\lfloor \frac{L}{\tau }\right\rfloor =N,$$
where 1≤b≤L and the FDispEn of each coarse-grained signal $y_{j}^{\left(\tau \right)}$ is calculated. The coarse-grained approximation signal y is mapped into $u=\left\{y_{1},y_{2},\cdots ,y_{N}\right\}$ from 0 to 1 as follows:(8)$$u_{j}^{\left(\tau \right)}=\frac{1}{\sigma \sqrt{2\pi }}\int_{-\infty }^{a_{j}^{\tau }}e^{\frac{-{\left(t-\mu \right)}^{2}}{2\sigma^{2}}}dt,$$
where μ and σ are the mean and standard deviation (±SD) of coarse-grained time series *y*, respectively. Then, each $y_{i}$ is assigned an integer from 1 to *c* as $z_{j}^{c}=round(c*y_{i}+0.5)$, where $z_{j}^{c}$ denotes the *j*-th member of the time series [45,46].

For an embedding dimension *m* and a time delay *d*, the time series $z_{i}^{m,c}$ can be defined as $z_{i}^{m,c}=\{z_{i}^{c},z_{i+d}^{c},\cdots ,z_{i+(m-1)d}^{c}$, where *i*={1,2,···,*N*-(*m*-1)*d*}. Each time series $z_{i}^{m,c}$ is mapped to a fluctuation-based dispersion pattern $\pi_{v_{0}v_{1}\cdots v_{m-1}}$, where $z_{i}^{c}=v_{0}$, $z_{i+d}^{c}=v_{1}$,…, $z_{i+(m-1)d}^{c}=v_{m-1}$. The number of possible dispersion patterns that can be assigned to $z_{i}^{m,c}$ is equal to ${\left(2c-1\right)}^{\left(m-1\right)}$.

For each $c^{m}$ potential dispersion pattern $\pi_{v_{0}v_{1}\cdots v_{m-1}}$, relative frequency is obtained as follows:(9)$$p(\pi_{v_{0}\cdots v_{m-1}})=\frac{\#\left\{i|i\leq N-(m-1)d,z_{i}^{m,c}\;has\;type\;\pi_{v_{0}\cdots v_{m-1}}\right\}}{N-(m-1)d},$$
where *#* refers to a cardinality. Therefore, the MFDE value is calculated by using Shannon’s definition of entropy as given below:(10)$$MFDE(y,m,c,d,\tau )=-\sum_{\pi =1}^{{\left(2c-1\right)}^{m-1}}p(\pi_{v_{0}v_{1}\cdots v_{m-1}})\ln p(\pi_{v_{0}v_{1}\cdots v_{m-1}}).$$

RCMFDE is the basis of MFDE calculation, which in turn is based on each time scale. The scale factor of the coarse-grained series τ is considered, with each τ corresponding to a different starting point in the process of coarse-grained. Then, from the screened series, the relative frequency of each fluctuation-based dispersion pattern is calculated. Finally, the RCMFDE value is defined as the Shannon entropy value of the average occurrence rate of the fluctuation-based dispersion pattern of those screened series. That is
(11)$$RCMFDE(x,m,c,d,\tau )=-\sum_{\pi =1}^{{\left(2c-1\right)}^{m-1}}\bar{p}(\pi_{v_{0}v_{1}\cdots v_{m-1}})\ln\bar{p}(\pi_{v_{0}v_{1}\cdots v_{m-1}}),$$
where $\bar{p}(\pi_{v_{0}v_{1}\cdots v_{m-1}})=\frac{1}{\tau }\sum_{k=1}^{\tau }p_{k}^{\left(\tau \right)}$ is the dispersion model in time series $x_{k}^{\left(\tau \right)}$ (1 ≤ *k* ≤ *τ*) of the relative frequency, *c* is mapped class, and *d* is the delay time.

#### 2.4.3. MMSWPE Extraction

For a signal $x=\left\{x_{1},x_{2},\cdots ,x_{L}\right\}$ with length L, a set of m-dimensional vectors formed as $V=\{x_{t+(j_{i}-1)l},x_{t+(j_{2}-1)l},\cdots ,x_{t+(j_{d-1}-1)l},x_{t+(j_{d}-1)l}\}$ of length o is given by sample $x_{j}$, with i and ranged from 1 to G, where G=N−(o−1)τ. Different samples have d! potential ordinal patterns, π, also known as “motifs”. The relative frequency is obtained as follows [50]:(12)$$p(\pi_{k})=\frac{\sum_{i=1}^{G}l_{v:type(v)=\pi_{k}}(V_{k}^{\left(\tau \right)})}{\sum_{i=1}^{G}l_{v:type(v)=\Pi }(V_{k}^{\left(\tau \right)})},$$
where $\Pi =\{\pi_{j}{\}}_{j=1}^{m!}$, $l_{v}$ denotes the indicator function of set A={d!}, which defined as $l_{v}=0$ if v∉A and $l_{v}=1$ if v∈A, $V_{k}^{\left(\tau \right)}$ stands for *k*-dimensional vectors in a specified length of time.

Therefore, PE is calculated by using Shannon’s definition of entropy as follows [51]:(13)$$PE(o,d,\tau )=-\sum_{\pi_{k}=1}^{\pi_{k}=d!}p(\pi_{k})\ln p(\pi_{k}).$$
For WPE, transformation Equation (12), the probability distribution of each “motif” is defined as follows:(14)$$p_{w}(\pi_{k})=\frac{\sum_{i=1}^{G}l_{v:type(v)=\pi_{k}}(V_{k}^{\left(\tau \right)})w_{i}}{\sum_{i=1}^{G}l_{v:type(v)=\Pi }(V_{k}^{\left(\tau \right)})w_{i}},$$
where $w_{i}$ is the weighted value for vector V and is calculated by the variance of each adjacent vector V, and the mean of the vector V is $\bar{V}$, thus, $w_{i}=\frac{1}{m}\sum_{q=1}^{m}[x_{i+(q-1)\tau }-\bar{V}{]}^{2}$. WPE is the extension of PE, and it saves useful amplitude information included in the signal, which is computed as
(15)$$WPE(o,d,\tau )=-\sum_{\pi_{k}=1}^{\pi_{k}=d!}p_{w}(\pi_{k})\ln p_{w}(\pi_{k}).$$

The MMSWPE method, which combines WPE with the multivariate multiscale method was proposed to extract the complexity of different signals more effectively. Therefore, MMSWPE, which represents microwave transmission signal complexity, is calculated as
(16)$$MMSWPE(o,d,\tau )=-\sum_{\pi_{k}=1}^{\pi_{k}=d!}{\bar{p}}_{w}(\pi_{k})\ln{\bar{p}}_{w}(\pi_{k}).$$

Therefore, we calculate RCMSE, RCMFDE, and MMSWPE by setting six parameters, namely, the embedding dimension *m*, mapping class *c*, delay time *d*, scale factor τ, tolerance *r*, and permutation entropy order number *o*. In this study, *c* and *d* are known to be 6 and 1, respectively and *m* is determined according to $c^{m}<L$, *L* = 2500, hence it is set as *m* = 4. τ = 6 can guarantee a satisfactory performance and will not have too many features that could influence the classification effect. r=0.15σ(x), σ(x) represents the standard deviation of the original time series *x*, and *r* is too large to cause information loss. As suggested by reports [51,52], when σ(x) is 1, the signal is most stable; thus *r* is set to 0.15. On the basis of numerous experiments and experience, the performance is the best when *o* is set to 3 in this paper [50].

### 2.5. Feature Selection and Classification

#### 2.5.1. Feature Selection

Each time series was calculated to obtain 5 × (6 + 6 + 6) = 5 × 18 features, and each subject obtained a 40 × 4 × 4 × 5 × 18 five-dimensional feature matrix. Then, 24 combinations of subjects obtained a 24 × 40 × 4 × 4 × 5 × 18 six-dimensional feature matrix, adjusting the dimension to a 15,360 × 90 feature matrix. A six-dimensional feature matrix with a combination of 24 × 40 × 4 × 4 × 5 × 18 was obtained by 24 subjects and we adjusted the dimensions to a 15,360 × 90 feature matrix. The number and dimensions of these features are large and the feature dataset has too many dimensions or redundant features, which not only poses challenges to the classifier design and training but also worsens the classification effect and considerably increases computational complexity because of the possible “dimension disaster” [53]. Therefore, it is necessary to select the feature matrix. In the feature selection process, principal component analysis (PCA) can only measure the linear relationship between variables but not the nonlinear relationship between features [54]. The purpose of the minimal-redundancy-maximal-relevance (mRMR) criterion based on mutual information is to maximize the dependence between variables, which includes the calculation of multivariate joint probability. The process is complicated and very difficult [55]. If the mutual information method is combined with the PCA algorithm, the estimation of multivariate density, which is difficult to achieve in the process of dependency maximization, can be avoided in feature selection. Therefore, this paper proposes a mutual information-principal component analysis (MIPCA) feature selection method based on mutual information. The method preserves the probability distribution of features in PCA, the self-information between features, and the mutual information between features.

For a feature dataset $F^{R\times D}$, where feature $F_{i}|f_{i}^{M},M=1,2,\cdots ,R$ and feature $F_{j}|f_{j}^{N},N=1,2,\cdots ,R$. If their probability density functions and joint probability density functions are $p(f_{i})$, $p(f_{j})$, and $p(f_{i},f_{j})$, respectively, their mutual information can be expressed as [55]:(17)$$I(F_{i},F_{j})=\sum_{M}\sum_{N}p(f_{i},f_{j})\log\frac{p(f_{i},f_{j})}{p(f_{i})p(f_{j})}.$$

Then, the principal component matrix based on mutual information can be calculated using the following formula:(18)$$B^{\prime }\sum_{\;I(F_{i},F_{j})}B=\Lambda ,$$
where B is the matrix composed of the feature matrix $\left\{\beta_{1},\beta_{2},\cdots ,\beta_{\kappa }\right\}$, and the corresponding feature vectors are orthogonal in pairs and are orthogonal matrices, Λ is a matrix composed of the feature vector $\left\{\gamma_{1},\gamma_{2},\cdots ,\gamma_{\kappa }\right\}$, which is a diagonal matrix, and $\sum_{\;I(F_{i},F_{j})}$ is the mutual information matrix, which is a symmetric matrix of non-negative real numbers.

The principal component information based on mutual information can be expressed as follows:(19)$$P_{\kappa }=B^{\prime }f_{\kappa^{\prime }},$$
where the variables are orthogonal to each other. Dimension $D_{\kappa }$ of the principal component is determined next. The contribution rate of the principal component of MIPCA is defined as $C_{\kappa }$, which is the ratio of the single principal component to total principal component information, that is,
(20)$$C_{\kappa }=\mu_{\kappa }/\sum_{\kappa =1}^{N}\mu_{\kappa },$$
where $\mu_{\kappa }$ is the characteristic value of the κ-th largest mutual information matrix $\sum_{\;I(F_{i},F_{j})}$, which represents the information amount of the principal component κ.

The contribution rate of the principal component in dimension $d_{\kappa }$ is the sum of the contribution rates of the previous $d_{\kappa }$, that is,
(21)$$\delta_{\kappa }=\sum_{i=1}^{\kappa }C_{i}.$$

The first $D_{\kappa }$ principal components with a contribution rate of 85–95% were selected as the new feature.

#### 2.5.2. Classification and Performance Evaluation

MIPCA was used for feature selection and dimensionality reduction of feature dataset. The features of the first 20 with a contribution rate of ≥ 85% were selected as feature subsets for training and classification. Leave-one-out Cross Validation (LOOCV) was used to capture the optimal classification model, leaving all data of one subject at a time as the test dataset and the rest as the training dataset. Finally, a generalized classifier of non-specific individuals was generated. The RF classifier was then used to complete multi-class task recognition. For RF, the random selection of training datasets and feature subsets depends on LOOCV to construct, test, and validate the decision tree. The C4.5 decision tree was selected and the leaf node containing the minimum sample number minleaf was 36 to generate the optimal decision tree. The test results were evaluated, and the optimal classification selected by voting to generate the optimal RF classifier. Then, the optimal classifier was used to categorize the classification performance of the test dataset. The evaluation indicators are true positive (TP), true negative (TN), false positive (FP), and false negative (FN). The performance of the RF classifier is evaluated using accuracy and precision. The calculation method is as follows:(22)$$A\mathrm{cc}uracy=\frac{TP+TN}{TP+FN+TN+FP}\times 100\%,$$
(23)$$P\mathrm{recision}=\frac{TP}{TP+FP}\times 100\%.$$

## 3. Results

### 3.1. Evaluation of Improvement of Brain Function Sites and Sleep Quality by Microwave

The microwave modulation frequency (20 Hz) acts on the active area of the brain, changes the firing frequency of its functional sites, inhibits the firing activity of the awakening nerve cells, increases the tendency of brain activity to calm down, and improves sleep quality. To test the effect of microwaves on sleep, patients filled out a survey rating form when they woke up after each true machine test, and their sleep quality was rated on a scale of 0 to 10. The higher the score, the more satisfied with the sleep improvement. Figure 3 shows the statistics of the sleep quality satisfaction rating scale of the subjects under the condition of a true machine. It shows that the subjects’ satisfaction with the effect of microwave treatment is above eight points, representing "relatively satisfied", indicating that the effect of microwave on the functional sites of the brain really improves the sleep effect of the patients with sleep disorders.

To objectively evaluate the improvement of sleep quality, we used PSG records to identify each subject’s sleep condition throughout the night. Figure 4 shows that in the first test, the neuralgia subjects who did not receive microwave treatment (false machine) had less than 15% sleep time. After microwave treatment (true machine), the patients’ proportion of sleep time continued to improve, maintaining above 80% overall, and the insomnia rate decreased gradually. After four tests, the average sleep time was also more than 70% without the assistance of microwave, indicating that a significantly better improvement of sleep quality due to the microwave. A comparison of the paired *t*-test results of the true and false machine in four tests, we find that as the number of treatments increased, the difference between the true and the false machine tests became increasingly smaller. Hence, we can conclude that microwaves can achieve good results within a short time and taper off the use of microwaves to help people to sleep, which is a safer and more reliable option than the long-term use of drug therapy. Figure 5 shows the average SOL of patients receiving treatment on the true machine and the false machine. After the addition of electromagnetic wave therapy, the sleep time of patients were improved significantly and thus, the SOL could be advanced and SMR increased. As the number of times microwaves aid sleep increases, SOL continues to decline as shown in Figure 5. The paired *t*-test of the true and the false machines showed that the *p*-value was less than 0.05 or 0.01, indicating that the true machine improved sleep quality significantly better than the false machine.

Figure 6 shows that when the subjects were in the true machine environment, in a EOG channel, the power spectral density variation of brain activity was significantly lower than that in the false machine environment. The explanation for this finding is that the microwave does change the firing frequency of brain function sites, causes the degree of activity of the cerebrum to decrease, thereby facilitating a person to fall asleep.

Under microwave regulation, the frequency band energy percentage of each sleep stage was compared, as shown in Table 1. In each sleep phase, the highest frequency band energy percentage was in the delta band, while the variations in other frequency bands were not consistent. Delta had the highest proportion in the SWS phase, and lowest when it was in REM. As sleep progresses, the insomnia rate gradually decreases, and the energy ratio in the theta, alpha, and beta bands becomes lower and lower.

Table 2 shows the proportion of each sleep stage in TIB and the *p*-value of the paired *t*-test for the true and the false machine tests. As can be seen from Table 2, there was a significant statistical difference between the true and false machine tests in the W and LS stages, indicating that microwave treatment could significantly reduce W and increase LS. However, the improvement of SWS stage by microwave treatment was not significant, and there was no improvement in REM stage. In the false machine test, the highest proportion of W stage was 58.72%, and the TST was only 41.28%, indicating that the patient was in a state of severe insomnia. In the true machine test, the W stage dropped to 18.44%, and the TST accounted for 81.56%. In general, microwaves significantly improved sleep quality in people with sleep disorders.

### 3.2. Multiscale Entropy Extraction

RCMSE, RCMFDE, and MMSWPE were extracted from the wavelet packet coefficients corresponding to the frequency band of each sleep stage: A(5,0), D(5,1), D(5,3), D(5,2) and D(3,1), respectively. The calculated mean entropy value is shown in Figure 7. By comparing the entropy at different stages, the following results are obtained; for coefficient of A(5,0), the entropy of LS period is the largest, indicating that LS in A(5,0) had the largest complexity. However, in D(5,1), D(5,3), and D(5,2), no uniform distinction between the corresponding entropy values in each stage was observed, thereby indicating that different entropy values in the wavelet packet coefficients will have different complexities. For D(3,1), its frequency band only appears in stage W and entropy is higher in stage W, indicating that the brain shows strong activity at this time. In general, for the coefficient of the low-frequency band, the shallower the sleep, the higher the entropy and the higher the complexity. Conversely, the higher the wavelet packet coefficient in the high-frequency band, the higher the entropy of deeper sleep. Figure 7 shows the significant differences in the entropy of different sleep stages. Each stage can be distinguished from other stages.

Figure 8 shows the average results calculated for RCMSE, RCMFDE, and MMSWPE in each sleep stage. The results show that the entropy value of stage W is the largest whereas that of LS and SWS decreased successively and SWS reached the lowest. REM increased and was close to LS, indicating that brain activity was the most active during the W stage, accompanied by complex thinking activities. W also had the largest corresponding complexity. With the deepening of sleep, the activity of the cerebral cortex is weakened gradually and brain firing activity is reduced, leading to the reduction of system complexity, that is, the complexity of the deep sleep stage is the lowest. During REM, however, because of the uncertainty as to brain activity, the degree of activity and complexity fluctuates but the entropy value hovers around LS or is slightly higher than LS.

### 3.3. Classified Evaluation 

The entropy features were obtained from data of each stage through WPT decomposition and in the selected wavelet packet coefficients. The features selected by MIPCA were combined into a new feature dataset and the RF classifier was constructed according to the LOOCV method, that is, the data of one person were selected randomly as the test dataset and the rest as the training dataset. The test dataset is classified, and the results obtained are shown in Table 3. The average classification accuracy is 86.41%, among which LS and SWS had the highest recognition rate and REM had the lowest, possibly because REM tends to alias with LS, which affects the detection precision.

The four types of sleep stages of the 24 subjects were classified and compared separately to verify the feasibility of microwave recognition of sleep stages and treatment of sleep disorders. Figure 9 shows the accuracy and precision of 24 subjects being classified. Figure 9 shows the detection accuracy of all subjects is more than 70% and the average accuracy is more than 80%. Similarly, the precision of each subject was maintained between 70% and 90%, with an average of over 85%. This finding suggests that microwaves have the same function as EEG and can be used to detect various types of brain activity. It also provides further evidence that microwaves can detect brain activity, such as pain, sleep, epilepsy, and depression, and can be used to treat and alleviate brain diseases caused by these factors.

The performance of the proposed method is compared with several existing method-based techniques in Table 4 in terms of overall accuracy. Although these methods use different databases, such as EEG or EOG signals, they all use entropy features for classification. The proposed method has the best overall performance with an accuracy of 86.41%. The detection accuracy achieved by the proposed method is significantly higher than those of other techniques. Table 5 is a comparison of the overall accuracy performance of the proposed method with existing methods of other feature types. The proposed scheme outperforms the others in most of the cases. Thus, microwave detection could be a viable alternative to EEG in sleep stage classification.

## 4. Discussion

The current research aims to develop an integrated system of non-contact sleep stage detection and sleep disorder treatment for health monitoring. Based on the principle of microwave scattering, the system uses a microwave transceiver instead of scalp EEG contact to evaluate sleep staging. After four microwave modulation treatments, sleep efficiency, SMR, and SOL were improved considerably. The quality of sleep has also improved markedly. The classification of microwave transmission signals for testing sleep shows that the average detection accuracy of sleep stages exceeds 80%, indicating that microwaves can replace EEG in detecting brain activity. However, different from EEG, microwave can not only analyze and identify the spectrum of the signal quickly based on the detected brain activity signal through the calculation module but also send the modulation frequency to act on the brain functional sites, changing and adjusting the firing frequency of the brain functional sites to achieve the purpose of the treatment [16,17,18,19]. In view of the small amplitude of EEG, which is difficult to measure and because non-invasive and low-damage detection is a trend, the high frequency and strong penetration characteristics of the microwave makes it an important method for non-invasive detection [61].

In the sleep quality test, Figure 3, Figure 4, Figure 5 and Figure 6 shows the subjects’ sleep satisfaction scores, sleep efficiency, SMR, and SOL, which were calculated without considering the number of awakenings between sleep. All subjects experienced significant improvements in sleep quality after four microwave treatments. In particular, the efficiency of sleep increased to 80%, reaching the level of normal people. The SOL is reached much earlier, which means the rate of insomnia decreased gradually. The paired *t*-test (95% confidence level) with and without microwave therapy showed the differences in sleep efficiency and SOL decreased over time. By the fourth treatment, the difference was so close that some patients may have completed the treatment.

This paper also improved the traditional PSG technology using microwave detection without touching the scalp instead of EEG as a tool to obtain brain activity signals. The power spectral density variation with microwave regulation is lower than that without microwave regulation, indicating that brain activity decreased under microwave regulation. The direct reason may be because the subjects were drowsy or fell asleep during this process. Table 1 and Table 2 show the energy possession ratio and the relative TIB possession ratio for each sleep stage. The energy of slow waves during sleep is significantly lower than when they are awake. Based on the experimental results, as sleep deepened, the energy carried by the high-frequency component of the microwave signal decreased gradually [62]. In contrast, the energy of the low-frequency component increased gradually [63]. In terms of the proportion of energy of brain rhythm corresponding to each sleep stage, the proportion of the delta energy was the highest. However, the proportion of delta energy was higher 90% because of the decrease in brain activity. In general, neuronal activity in the brain decreases during sleep but does not disappear completely, with the beta wave energy percentage decreasing and the delta ratio increasing [64]. However, our results and the results of Sichari et al. [65] may also be explained by a common mechanism that leads to an increase in power spectral activity across the upper portion of the EEG spectrum including the slow waves and alpha frequency bands. The proportion of theta waves increased during REM and the proportion of delta waves was the smallest, indicating that during REM, brain activity was in a state of wakefulness and that the energy of theta and alpha waves began to increase [66]. When LS and SWS were in light and deep sleep, alpha and beta activities were the weakest, but the energy ratio of delta and theta waves increased significantly [67]. In addition, according to the paired *t*-test results with or without microwave modulation, significant differences in the proportion of TIB can be observed at each stage. Among them, W decreased by 49%, LS and SWS increased by 39% and 16%, respectively, which means that with the help of a microwave, sleep efficiency was improved, and sleep disturbance was alleviated.

In terms of extracting features, wavelet packets are more precise in extracting rhythm wave frequency bands, which can decompose high-frequency signals further such that each coefficient corresponds to a frequency band of brain rhythm, which is more conducive to feature extraction [68]. Figure 7 and Figure 8 show the RCMSE, RCMFDE, and MMSWPE extracted in each sleep stage. Among the features, those with the highest frequency of use are the ones in the time domain and frequency domain. However, because these features are linear signals that deal with non-stationary, nonlinear signals, such as microwave propagation, some information can be easily lost [69]. Therefore, the nonlinear feature may be a choice for identifying signal information [70,71]. In this paper, three nonlinear features of RCMSE, RCMFDE, and MMSWPE were extracted, which were effective in the sleep stage. Different entropies differ significantly in different wavelet packet coefficients, which can reflect fully the different stages of sleep [37,38,39,40]. Multiscale entropy can represent system dynamics features on multiple time scales and may reveal specific characteristics in different sleep stages, which is beneficial to the recognition of sleep types [72,73]. In general, neurons in the brain exhibit reduced activity during sleep but follow a specific pattern [74]. In fact, neither sleep nor wakefulness can determine whether a neuron system is inactive during sleep. Certain oscillatory behaviors and sometimes even some neurons are more active during sleep than during wakefulness [75]. Figure 7, Figure 8 and Figure 9 confirm this view, and the changes in different entropies in each rhythm are not necessarily consistent. However, in different sleep periods, the entropy value appears at a certain regularity because of the different degrees of nerve activity [76]. The higher the nerve activity, the higher the complexity of the signal, and the higher the value [45,46].

Before the recognition and classification of sleep stages, some feature dimensionality reduction or feature selection algorithms may be used to process the data to handle the large dimension of the extracted feature dataset, which affects the classification effect of the classifier [53]. For example, PCA and independent component analysis (ICA) can only measure the linear relationship between variables and may ignore important information [54]. The modified MIPCA retains the probability distribution of features in the PCA method and has self-information and mutual information between features to be expressed in mutual information, which is more favorable for retaining useful sleep staging data [55]. Table 3 and Figure 9 show that the recognition rate of all sleep stages was over 80%, the overall accuracy was 86.41, and the average recognition accuracy of each subject was over 70% when RF classification is used.

The proposed method uses only microwave recordings as the input signals. Compared to the conventional sleep scoring methods that require multiple physiological signals (EEG, EOG, and EMG), the microwave-based method has the advantage of reducing sleep disturbance caused by recording wires. It is especially helpful for home environments and clinical care. Although some methods based on EEG or EOG have been developed recently, Table 4 and Table 5 show that our method is superior to other methods, whether using similar entropy as a feature or other forms of the feature. Another advantage of our method is that the accuracy of each stage is more balanced and the classification accuracy reaches 86.41%, indicating that our method has higher reliability.

In summary, using microwaves instead of EEG tests could both eliminate the hassle of too many electrodes sticking to the skin and allow brain activity to be detected and identified in the same way as an EEG. In addition, certain microwave frequency modulation can improve the sleep quality of patients with sleep disorders. This effect may be due to microwaves changing the concentration of ions in the extracellular fluid of neurons in the functional sites of the brain during sleep, thereby altering the permittivity of neurons in the functional sites of the brain, resulting in a loss of energy in the brain’s electromagnetic field and the brain’s desire for drowsiness [16,17,18,19]. Therefore, the principle of microwave scattering can be used to design portable and inexpensive integrated microwave devices to detect and treat sleep disorders. For a variety of insomnia patients, including the elderly, children, women, microwaves can be used safely to improve their sleep without the dependence on neurological drugs. Moreover, it can obtain higher classification accuracy. Notably, the method is also is superior to other methods in the detection accuracy of sleep stages.

Our study has the following limitations: (1) We chose 6 GHz electromagnetic wave emission frequency and 20 Hz modulation frequency for the detection of sleep stages and the treatment of sleep disorders. However, we did not consider the comparison and validation of other frequencies because more reliable information on microwave detection and treatment of brain diseases is not available. (2) The comparison of multiple microwave frequencies to determine which microwave frequency is more favorable for detecting sleep activity is not considered. It is also not possible to identify which modulation band range is more favorable for regulating the firing behavior of the functional sites of sleep activity. Individual brains have certain differences in firing behavior of sleep and thus, a large error could occur in the detection and regulation using the same frequency standard. (3) For the sleep stage, wavelet packet technology with better robustness and multiscale entropy features are used to distinguish between the features of different sleep stages. Excessive consideration of a single type of nonlinear feature may affect the accuracy of the sleep stage. In view of these defects, our next research focus will be to consider and solve the problem.

## 5. Conclusions

In this paper, the principles of microwave scattering are used for the first time in the design of a microwave detection and treatment system. Following the problem of a popular sleep disorder, the microwave is used to modulate the brain functional sites and change the abnormal firing behavior of the activated area to improve and treat insomnia. This paper is also the first time that a microwave has been used to detect sleep brain activity. The proposed system is found to be capable of improving on the shortcomings of traditional PSG technology, replacing the inconvenient EEG contact connection and enabling non-contact microwave antenna to monitor sleep brain activity. After extracting the multiscale entropy features of microwave transmission signals and evaluating the performance of RF classifier, the average classification accuracy was 86.41%. Hence, the proposed system is proven to be successful despite many shortcomings. In the future, we will focus on the defects in the current research and improve the research methods further to achieve higher classification accuracy and treatment effect. Finally, we designed a more effective and convenient integrated microwave system that can be used widely in the daily detection and treatment of insomnia and could offer more effective treatments for other brain disorders, such as epilepsy, depression, and consciousness disorders.

## Figures and Tables

**Figure 1 entropy-22-00347-f001:**
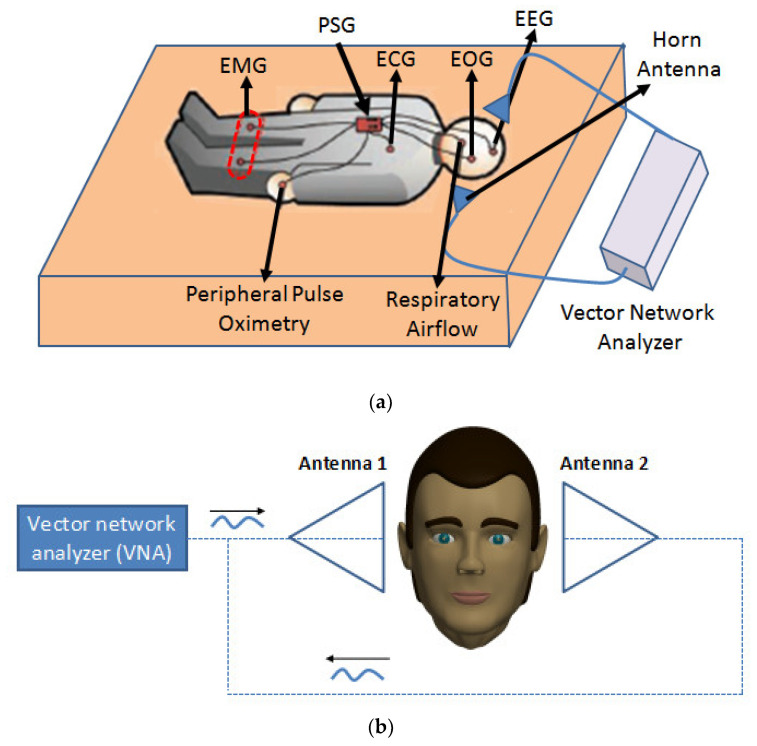
Schematic diagram of experimental paradigm: (**a**) sleep disorder treatment and sleep staging detection system; (**b**) schematic diagram of microwave emission and recording.

**Figure 2 entropy-22-00347-f002:**
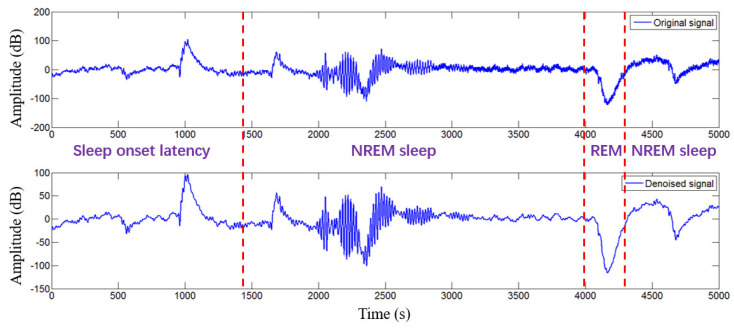
Comparison of pre-denoising and post-denoising of a microwave transmission signals in the first 5000 s sleep test. The signal was digitally bandpass filtered using a fourth-order Butterworth filter between 0.5 and 150 Hz. The forward and backward filtering is used to reduce the phase distortion. Then, the time series was digitally filtered using the hamming window FIR bandpass filters of order 200, with cutoff frequencies of 0.5 Hz and 40 Hz, which are usually used to analyze brain activity.

**Figure 3 entropy-22-00347-f003:**
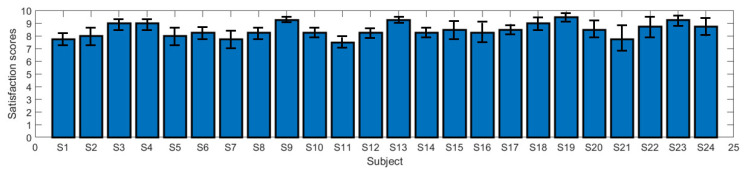
Bar chart of mean satisfaction scores of all subjects.

**Figure 4 entropy-22-00347-f004:**
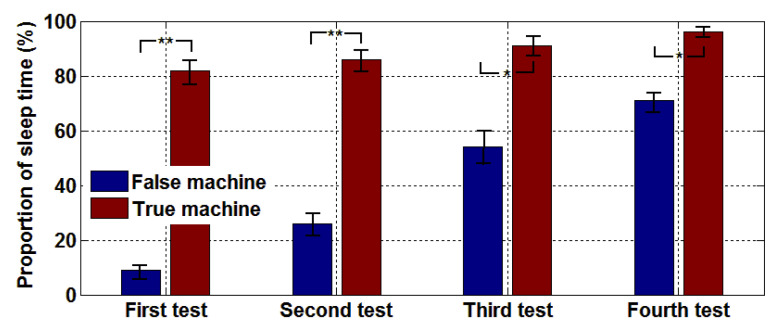
Comparison of sleep efficiency of four tests, * *p* ≤ 0.05, ** *p* ≤ 0.01.

**Figure 5 entropy-22-00347-f005:**
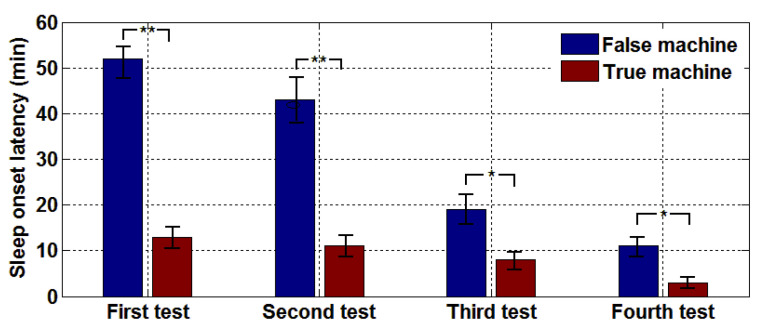
SOL comparison chart for four tests, * *p* ≤ 0.05, ** *p* ≤ 0.01.

**Figure 6 entropy-22-00347-f006:**
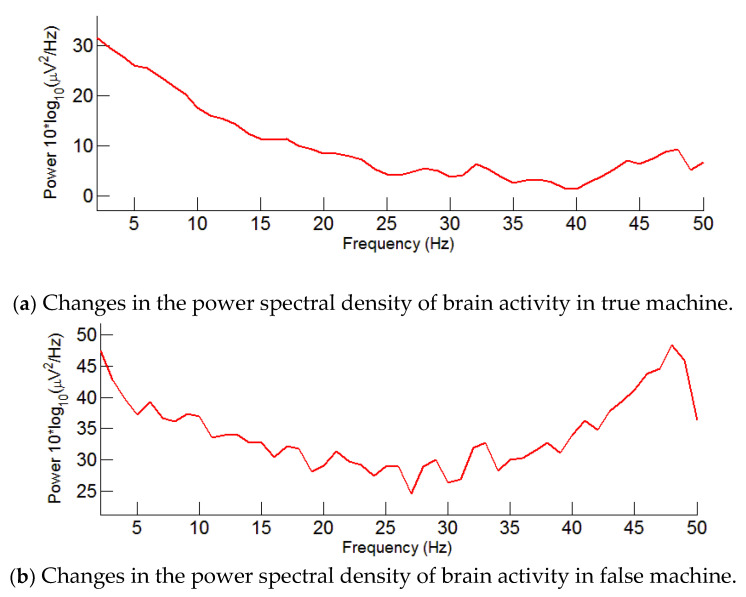
Variation of the power spectral density of brain activity under the modulation of a false machine and a true machine.

**Figure 7 entropy-22-00347-f007:**
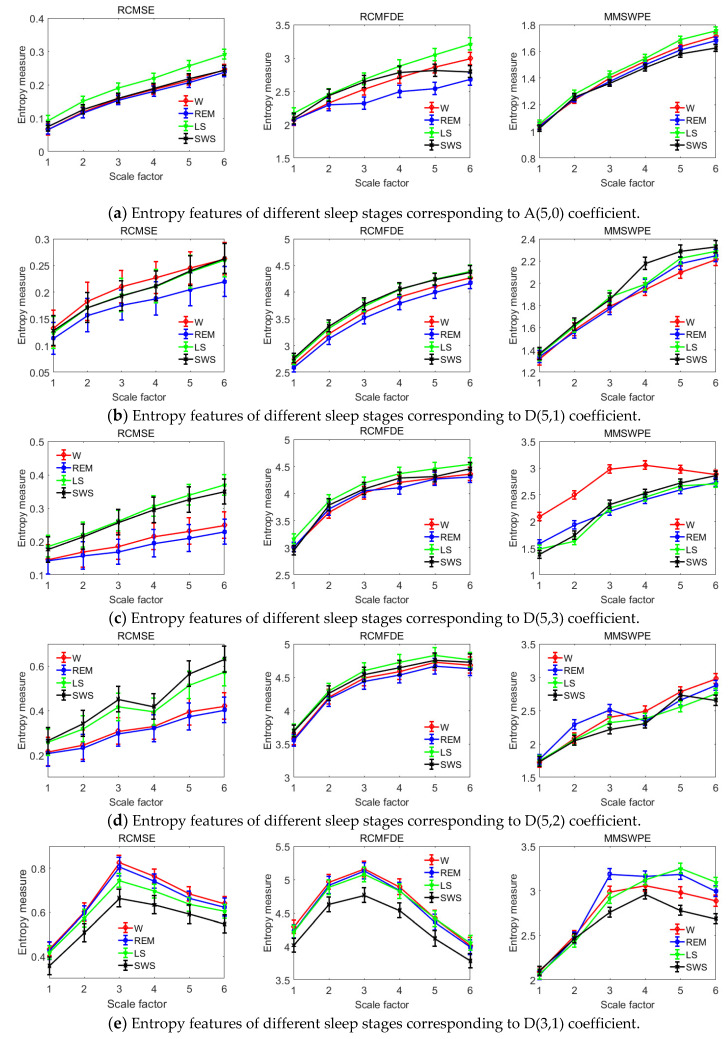
Mean measurements of RCMSE, RCMFDE, and MMSWPE for each sleep stage at different wavelet packet coefficients.

**Figure 8 entropy-22-00347-f008:**
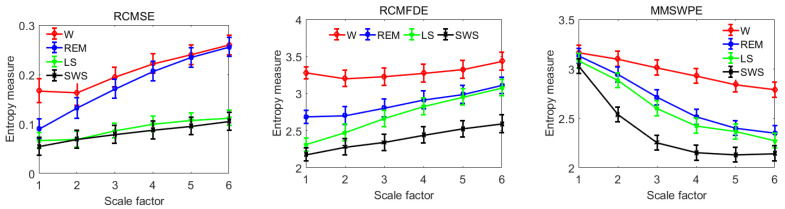
Mean measurements of RCMSE, RCMFDE, and MMSWPE at various stages of sleep.

**Figure 9 entropy-22-00347-f009:**
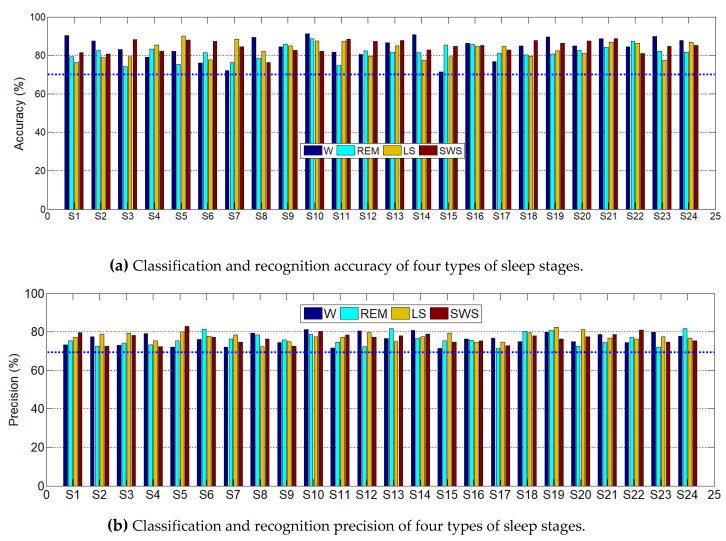
Mean classification precision and accuracy of the four types of sleep stages of the 24 subjects.

**Table 1 entropy-22-00347-t001:** Frequency band energy percentage of different sleep stages in the case with microwave regulation.

Sleep Stage	Delta	Theta	Alpha	Beta
W	69.23 ± 9.32	10.17 ± 2.34	3.35 ± 1.78	17.25 ± 3.78
REM	64.41 ± 8.73	20.19 ± 4.31	6.43 ± 1.87	8.97 ± 2.74
LS	79.54 ± 6.27	12.37 ± 1.58	4.62 ± 1.03	3.47 ± 1.23
SWS	91.12 ± 2.18	5.26 ± 0.68	2.26 ± 0.27	1.36 ± 0.19

**Table 2 entropy-22-00347-t002:** Proportion of each sleep stage in TIB with and without microwave regulation.

Sleep Stage	True Machine (%)	False Machine (%)	Improve?	*p*
W	18.44 ± 1.37	58.72 ± 2.53	Yes	<0.01
REM	23.79 ± 1.78	23.22 ± 1.46	No	0.54
LS	46.25 ± 2.83	11.17 ± 1.83	Yes	<0.01
SWS	11.52± 1.57	6.89 ± 1.64	Yes	<0.05

Note: *p* < 0.05 (0.01) means the difference is statistically significant.

**Table 3 entropy-22-00347-t003:** Classification results of test set by random forest algorithm.

	W	REM	LS	SWS
W	804	58	13	11
REM	86	782	58	27
LS	22	106	862	56
SWS	48	14	27	866
Accuracy	83.72%	81.45%	89.76%	90.17%
Overall Accuracy	86.41%			

**Table 4 entropy-22-00347-t004:** Comparison of the proposed scheme with existing methods under different entropy features.

Authors	Features (Database)	Methods	Overall Accuracy (%)
Rodríguez-Sotelo et al. [56]	Entropy metrics (EEG)	J-means approach	81.00
Liang et al. [48]	Multiscale entropy (EOG)	Linear discriminant analysis	76.91
Rahman et al. [47]	Refined composite multiscale dispersion entropy (EOG)	Random under-sampling boosting	84.70
Tian et al. [49]	Multiscale entropy (EEG)	Support vector machine	85.60
Proposed	Multiple multiscale entropy (microwave)	Random forest	86.41

**Table 5 entropy-22-00347-t005:** Comparison of the proposed scheme with existing methods for other feature types.

Authors	Features (Database)	Methods	Overall Accuracy (%)
Zoubek et al. [57]	Relative powers (EEG, EMG, and EOG)	Bayes rule-based classifiers	71.00
Tagluk et al. [58]	Hybrid features (EEG, EMG, and EOG)	Artificial neural networks	74.70
Ronzhina et al. [59]	Hybrid features (EEG)	Artificial neural networks	81.55
Fraiwan et al. [60]	Power spectrum (EEG)	Random forest	84.00
Proposed	Multiple multiscale entropy (microwave)	Random forest	86.41

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
