# Peer review of "A Novel Microwave Treatment for Sleep Disorders and Classification of Sleep Stages Using Multi-Scale Entropy"

_entropy, 2020, doi:10.3390/e22030347_

Round 1

Reviewer 1 Report

This paper presents an interesting research topic that uses microwave signals in non-contact sleep stage detection and sleep disorder treatment for health monitoring. With the experimental results reported in this paper, the microwave signal can replace the inconvenient EEG contact connection to monitor sleep brain activity. The limitations of this paper are clearly presented and would be considered as future extensions of this research.

Comments and suggestions are as follows.

Abstract: rewording the first phrase to be a sentence: “To develop an integrated system of non-contact sleep stage detection and sleep disorder treatment for health monitoring.”

Introduction: The following paragraph is firstly introduced microwave scattering technique however, no references are included: “Microwave scattering technique is a non-contact test method to avoid the inconvenience and anxiety caused by electrode attachment to the scalp, and to avoid the contamination of brain activity by physiological signal artifacts (i.e. EMG, EOG). And depending on its ability to record brain activity, it can be used in the study of sleep stage.” More details for the use of microwave scattering technique in similar research areas need to be reviewed and compared with the use of biosignals (EEG, ECG, etc.)

The next paragraph introduces microwave modulation and detection technology, however, no references for those are included, and no experimental results in other research areas for those are reported and compared.

It is not clear why the refined composite multiscale sample entropy (RCMSE), refined composite multiscale fluctuation-based dispersion entropy (RCMFDE) and multivariate multiscale weighted permutation entropy (MMSWPE) are used as feature data in this paper. A brief review of these feature extraction methods should be included to let the readers know where they have been used and why they are selected for this research.

The paper also needs to present the difference between the use of microwave modulation and detection technology, RCMSE, RCMFDE, and MMSWPE in this article and other published research articles to see the contributions of this paper.

Author Response

Dear Reviewer,

Thank you for your careful reading of our manuscript. I found the your comments are quite helpful, those comments are all valuable for revising and improving our paper, as well as the important guiding significance to our researches. We have carefully studied these comments and corrected them point-by-point. Furthermore, we have had the manuscript polished with a professional assistance in writing, and the sentences that have been modified are also marked in red. Thank you and the review again for your help!

Yours sincerely,

All authors

Reviewer 2 Report

The paper presents a novel microwave-based treatment for sleep disorders and also a sleep stage classification method using entropy. The paper is mostly well written, it has several strong points, such as:
1) the topic is interesting and has practical relevance,
2) the background and mathematical methods are adequately presented,
3) the conclusions and main messages of the paper are supported with experimental data.
For these reasons I can generally support the publication of the paper. However there are several mostly minor points which should be addressed prior to that:

  • Perhaps the only one major shortcoming of the paper is the often bad use of English. There are lots of mostly small grammar and wording mistakes, which altogether render the paper rather hard to read at some places. A detailed review by a native speaker is recommended. Some examples: The first sentence of the abstract does not contain a predicament. Probably half of the sentence is missing. Or: “Animals in the process of sleep mainly to anabolism, to restore physical strength and energy.” Also small grammar errors: “…sleep disorders is one of the…” -> are. Etc.
  • There are two distinct parts of the experiment, one is the detection and classification of sleep stages, the second is the treatment, both with the use of microwave. Based on Fig. 1 and the accompanying description on page 4 it is not clear whether “false machine” and “true machine” refer to both detection and therapy, or therapy only. If sleep detection based on the microwave data was also missing during the false machine tests, then how did the authors objectively measure sleep time and compare false and true machine tests based on this (Fig. 5)? Please define false and true machine tests, and indicate this in the text.
  • In Fig 2, the X axis should be in time instead of sample no. Also, please indicate in the caption the type and cut frequency of the filter used. Also, it would be informative if you could mark and label the different phases in the graphs.
  • On page 10 is real computer the same as true machine? Please be consistent with the naming.
  • The meaning of Fig. 3 is not clear. What was the precisely the question for this survey? How would they rate the sleep itself (as suggested by the experimental section)? Or how would they rate the improvement of their sleep due to the microwave treatment? (In the text here the authors state: “It shows that the subjects' satisfaction with the effect of microwave treatment is basically above 8 points…”) These two questions are not the same, and the second is moreover a guided question. Also, if the bars represent mean scores, please give error bars as well.
  • Page 11: “…the SOL continues to advance…” The sleep onset latency decreases in the function of microwave treatment steps, based on Fig.
  • In Fig. 6: the captions say “changes in spectrum” and “variation”, but all we see is one spectrum for true machine and one for false. Spectra represent variation in power in function of frequency, but we do not see here the “variation in spectrum”.
  • In table 2 I do not understand the “Improve?” column. Both cases (true, false) sum up to a total of 100%. It is not possible for all of the stages to show improvement. If this column refers to absolute times and not just proportions then please indicate because in its current form the table is misleading.
  • In Fig. 9 the accuracy and precision graphs seem to be identical. Are the authors sure they are correct?
  • Please revise the abstract and the conclusions: clearly indicate the most important findings and give precise numerical results. E.g. “The overall classification accuracy of the four sleep stages was more than 70%, with an average accuracy of over 80%”.” Why not use the exact value? Also, I would suggest trying to refrain from emphasizing the shortcomings in the conclusions (especially without being specific). They are already discussed at the end of the discussion section.

Author Response

(The authors gave the same response as above.)

Round 2

Reviewer 1 Report

The paper has been revised and improved significantly. It is good for publication with this journal. 

Reviewer 2 Report

I reckon that the authors undertook the effort to correct their paper. Its quality has improved significantly, thus I can support its publication in the journal.